# Alpha-Synuclein and the Endolysosomal System in Parkinson’s Disease: Guilty by Association

**DOI:** 10.3390/biom11091333

**Published:** 2021-09-09

**Authors:** Maxime Teixeira, Razan Sheta, Walid Idi, Abid Oueslati

**Affiliations:** 1CHU de Québec Research Center, Axe Neurosciences, Quebec City, QC G1V 4G2, Canada; maxime.teixeira@crchudequebec.ulaval.ca (M.T.); razan.sheta@crchudequebec.ulaval.ca (R.S.); walid.idi@crchudequebec.ulaval.ca (W.I.); 2Department of Molecular Medicine, Faculty of Medicine, Université Laval, Quebec City, QC G1V 0A6, Canada

**Keywords:** alpha-synuclein, endolysosomal system, vesicles, aggregation, trafficking, Parkinson’s disease

## Abstract

Abnormal accumulation of the protein α- synuclein (α-syn) into proteinaceous inclusions called Lewy bodies (LB) is the neuropathological hallmark of Parkinson’s disease (PD) and related disorders. Interestingly, a growing body of evidence suggests that LB are also composed of other cellular components such as cellular membrane fragments and vesicular structures, suggesting that dysfunction of the endolysosomal system might also play a role in LB formation and neuronal degeneration. Yet the link between α-syn aggregation and the endolysosomal system disruption is not fully elucidated. In this review, we discuss the potential interaction between α-syn and the endolysosomal system and its impact on PD pathogenesis. We propose that the accumulation of monomeric and aggregated α-syn disrupt vesicles trafficking, docking, and recycling, leading to the impairment of the endolysosomal system, notably the autophagy-lysosomal degradation pathway. Reciprocally, PD-linked mutations in key endosomal/lysosomal machinery genes (LRRK2, GBA, ATP13A2) also contribute to increasing α-syn aggregation and LB formation. Altogether, these observations suggest a potential synergistic role of α-syn and the endolysosomal system in PD pathogenesis and represent a viable target for the development of disease-modifying treatment for PD and related disorders.

## 1. Introduction

Accumulation of proteinaceous intraneuronal inclusions in the brain, also referred to as Lewy bodies (LBs), is the main neuropathological hallmark of Parkinson’s disease (PD) and related disorders [1,2]. These inclusions are mainly constituted of aggregated α-synuclein (α-syn) assembled in well-ordered fibrils [3,4,5]. α-Syn is a 140-amino acid presynaptic protein comprised of three domains: the N-terminal region (AA 1–60) which plays a role in modulating α-syn interactions with membranes; a central domain referred to as the non-Aβ component of AD amyloid (NAC) region (AA 61–95) very rich in hydrophobic amino acids essential for α-syn aggregation; and the C-terminal acidic carboxy-terminal tail (AA 96–140) implicated in regulating α-syn nuclear localization and fibrillization, as well as interactions with metals, small molecules and other proteins [1,6,7].

Although α-syn is commonly considered as the main component of LBs, new research has reported that these abnormal proteinaceous aggregates are made of more than just accumulated insoluble proteins, but are also composed of other cellular components such as cellular membrane fragments, vesicular structures, and dystrophic organelles [8,9,10]. These studies have helped better refine previous findings reporting the presence of lipids and other membranous fragments within LBs [11,12,13]. However, this observation may not be too surprising considering some of the physiological functions of α-syn and its role in the trafficking of lipid vesicles in the brain [14].

Physiologically, α-syn is found in its soluble form, associated with neuronal membranes regulating the recycling of synaptic vesicles (SVs) [15,16]. The process of α-syn and vesicle interaction has been shown to be an essential step in the regulation of functional processes, including those related to endoplasmic reticulum-to-Golgi vesicle trafficking [17] and recycling of the SVs during neuronal communication [18]. This described physiological role relates to how α-syn promotes the SNARE-complex assembly, allowing it to play a neuroprotective role since α-syn knockout (KO) considerably reduces the synaptic vesicle pool while also impairing the mitochondria and microglia [18,19,20,21,22,23]. As an abundant presynaptic protein, a number of studies point to its biological role in interacting with SVs modulating vesicle recycling and assembly, in addition to a role in regulating SV exocytosis and/or endocytosis [18]. α-Syn regulates the trafficking of SVs in the distal reserve pool to control the number of vesicles docked at the synapses for neurotransmitter release [24]. Indeed, studies have linked a direct physiological role of α-syn in binding to and inducing SVs clustering as observed both in vitro and in vivo [17,25,26]. Structural studies of α-syn have revealed that it is capable of adopting various conformations, allowing it to confer strong affinity to curved membranes, and these structural studies have suggested a physiological function of α-syn in SVs trafficking [26,27]. Furthermore, a study by Diao et al., showed that α-syn induces vesicle clustering through its interactions with VAMP2 and other negatively charged lipids through its lipid-binding domain [25]. Another study describes a phenomenon referred to as the “double-anchor” mechanism which further demonstrates α-syn interactions with SVs [28]. This study identified that specific residues of the N-terminus region, and central segment, function as membrane anchors, by which α-syn can bind two different vesicles, therefore prompting vesicle clustering [28].

On the other hand, aggregation and LBs formation alter α-syn physiological functions, resulting in the deregulation of vesicles homeostasis, leading to the manifestation of PD and related disorders [29]. Mutations and duplication/triplication of *SNC*, the gene coding for α-syn, have been associated with familial forms of PD, as well as increased α-syn aggregation propensity in experimental models [1]. Converging findings suggest that these α-syn fibrillar forms can also spread in a prion-like manner through synaptically connected brain regions, thus leading to PD disease progression [30,31,32,33,34]. This spreading might involve the vesicular system where exosomes have been proposed as the cargo for intercellular α-syn trafficking [35,36].

All these observations are pointing to the importance of the interaction between α-syn and vesicular trafficking, which also suggests that the impairment of the endolysosomal system might play an important role in the pathophysiological processes underlying PD and related α-synucleinopathies [36,37]. In this review, we will discuss work that highlights the interaction between α-syn and the main components of the endolysosomal system, in addition to dissecting recent findings on the role of these components in neuronal survival and more importantly, how α-syn impairs their function in PD.

## 2. The Endolysosomal System Is Impaired in Parkinson’s Disease

### 2.1. Introduction to the Endolysosmal System

The endolysosomal system is a very dynamic and complex system that mediates cellular connection with the extracellular environment through endocytosis, neutralizes pathogenic cargoes through phagocytosis, enhances the cellular proteolysis through autophagy, and maintains cellular homeostasis through endosomal sorting and trafficking of various proteins inside the cell (Figure 1) [38].

The endosomal pathway starts with clathrin-mediated endocytosis and vesicle uncoating, followed by fusion with early endosomes [39]. These early endosomes facilitate the uncoupling of ligands and membrane-bound receptors for recycling, while playing a role as a sorting station, directing cargo towards multiple destinations: (i) directly to the cell surface through recycling endosomes, (ii) to the lysosomes through maturation-mediated by endosomal sorting complexes required for transport (ESCRT) directed for degradation or (iii) to the *trans*-Golgi network (TGN) for retrograde sorting (Figure 1A) [39]. The key regulators of these pathways are the Rab GTPases (Rabs) proteins, which are implicated in maintaining specific interactions with other effector proteins such as coat proteins (clathrin-AP1, COPI, and AP3), motor proteins (Dynein-Dynactin, KIF5B, KIF1A, and KIF13A), and tethering complexes such as EEA-1 and SNAREs proteins (Figure 1B) [40,41,42].

Rab5, the early endosomal Rab GTPase involved in clathrin-mediated endocytosis and endosomal maturation, is one of the key regulators of synaptic vesicles trafficking [43]. In fact, these sorting endosomes can redirect their cargoes towards the cell surface, with the aid of other Rab family members, Rab4 and Rab11, two key regulators of this pathway [38,44]. The early endosomes targeted to the ESCRT pathway (degradation process) undergo a Rab5 to Rab7 conversion and mature into late endosomes (Figure 1A) [38,39]. In these endosomes the luminal pH decrease, while lysosomal enzymes concentration increases as the endosomes come closer to the nucleus [38,39]. During this maturation process, ESCRT allows for the ubiquitin-dependent sorting of cargo hydrolases from the outer membrane of early endosomes into intraluminal vesicles (ILV), forming multivesicular bodies (MVBs) (Figure 1A) [38]. Rab7, another member of the Rab GTPase mainly found on late endosomes’ membrane, acts as a key regulator of MVBs maturation from early endosomes; and its presence is also essential for the fusion of MVBs with lysosomes [44]. The late endosomes then deliver their luminal contents, by forming tubular extensions fusing with lysosomes, to initiate the proteolytic degradation of cargo (Figure 1A) [45]. Compelling evidence has reported that alterations of these proteins can directly affect neuronal activities in both physiology and disease [37].

The endolysosomal system also interacts with the autophagy and phagocytosis pathways. In fact, phagocytosis results in the fusion of phagosomes containing the cargo that are then subject to degradation through late endosomes or lysosomes (Figure 1A) [39]. The Autophagy-Lysosome Pathway (ALP) is commonly reported as two of the major protein degradation systems (with the ubiquitin-proteasome pathway) affected in synucleinopathies. ALP is the mechanism involved in the degradation of long-lived proteins, cellular components, and organelles via lysosomal fusion and function [46]. ALP is divided into three distinct pathways depending on the way substrates are destined to reach the lysosomal: macroautophagy, microautophagy, and chaperone-mediated autophagy (CMA) (Figure 1A) [47,48]. The macroautophagy pathway involves the formation of autophagosomes that sequester substrates—such as aggregated proteins or organelles—prior to fusing with lysosomes [49]. These processes, regulated by the complex ULK1 mandatory for autophagosome biogenesis, induce the formation of a double-membrane autophagosome around components tagged for degradation, further undergoing a series of maturation steps before fusing with late endosomes or lysosomes (Figure 1A) [39]. Microautophagy appends via the direct absorption of small untagged components by the lysosomes [39]. CMA functions through a process that involves the recognition of a specific Lys-Phe-Glu-Arg-Gln (KFERQ) peptide motif [50,51]. In fact, the process starts with a chaperone heat shock cognate protein of 70 kDa (hsc70) identifying a protein, which then allows for its translocation into a lysosomal membrane via the lysosomal-associated membrane protein (LAMP-1/2A). The protein is then disassembled into monomers by the lysosome-associated hsc70 (lys-hsc70) chaperone protein (Figure 1A) [52,53].

There has been great progress in understanding the endolysosomal pathway, its biology, and its role in health and disease. Over the years, more and more dysfunctions of the endolysosomal system have been associated with neurodegenerative diseases. Today, the understanding of how each component of this pathway can contribute or impair essential mechanisms of protein aggregation is a question that many studies have tried to resolve.

### 2.2. Endolysosomal Impairment in Parkinson’s Disease and Related Disorders

Converging lines of evidence from clinical observations and experimental studies suggest that dysfunction of the endolysosomal system may trigger neurodegenerative disorders, including PD. Genetic studies revealed a correlation between mutations in several genes encoding for endolysosomal proteins and PD pathogenesis. For instance, several PD-linked mutations are associated with lysosomal proteins, clathrin-mediated endocytosis, and retromer trafficking. The most common genetic mutations in lysosomal proteins, listed as a common genetic risk of PD, are those linked to the gene encoding for glucocerebrosidase (GBA) [54]. This mutation is associated with a reduction of the GBA enzymatic activity, and impairment of the autophagic function [55,56,57]. Another mutation in the gene *ATP13A2* linked to late endosomal/lysosomal protein deregulation has been identified in patients with early-onset PD [58]. Moreover, whole-exome sequencing of families with PD has identified mutations in the endolysosomal proteins required for the clathrin uncoating and vesicle endocytosis encoded by *DNAJC13*, *DNAJC6*, and *SYNJ1* [59,60,61,62]. More recently, mutations in vacuolar protein sorting-associated protein 35 (VPS35) and vacuolar protein sorting-associated protein 26A (VPS26A), two components of the retromer complex, have been associated with familial and sporadic forms of PD [63,64]. Other mutations in trafficking proteins were also found to be associated with PD, including mutations identified in the gene encoding for Rab7L1 in the *PARK16* locus linked to increasing the risk of PD [65,66]. Similar to α-syn gene mutations, mutations in *LRRK2* have been associated with monogenic forms of autosomal dominant PD [67]. Recent studies have shown that the overexpression of *LRRK2* mutants affects the interaction of Rabs with their effectors, which as a consequence can alter the endosome-lysosomal pathway [68,69]. Taken together, these genetic studies suggest that mutations in multiple proteins throughout the endolysosomal system play a central role in PD pathogenesis.

Of note, although several genetic factors clearly play a role in the impairment of the endolysosomal system in PD, α-syn seems to be implicated in the dysfunction of the major pathways of the endolysosomal system.

## 3. Interactions between α-Syn and the Endolysosomal System in Parkinson’s Disease

### 3.1. Overexpression of Monomeric α-Syn Disrupts Vesicle Trafficking

Interactions between the monomeric soluble form of α-syn and vesicles of the endolysosomal system have been well investigated, especially since the accumulation of α-syn has been reported to disrupt protein degradation and vesicle trafficking (Figure 1) [70]. Moreover, the overexpression of monomeric α-syn in yeast and in dopaminergic neurons has been associated with inclusion formation, impairment of Rab1 function, and the endoplasmic reticulum to Golgi trafficking [71]. In the secretory pathway of yeast cells, the accumulation of monomeric α-syn has been reported to disrupt early membrane trafficking by inhibiting vesicle docking, resulting in the accumulation of transport vesicles in vivo [17]. Moreover, yeast putative transmembrane protein 1 (Ytp1), the Rab protein ortholog in yeast, has been shown to predominantly colocalize with α-syn inclusions [17]. Additionally, certain Rab proteins, namely Rab3a, Rab8, and Rab11a, are able to directly interact with α-syn in cell culture and in vivo, reducing α-syn-induced toxicity [72,73,74,75]. 

In addition to the yeast model, other cellular models confirmed α-syn co-localization with transport vesicles in epithelial and neuroblastoma cells [76]. In fact, such in vitro assays have pinpointed for a specific localization of monomeric α-syn with overexpressed Rabs such as Rab5a, Rab7, and Rab11a, which are key effectors of early endosomes, late endosomes, and recycling endosomes respectively [76]. Similarly, the overexpression of monomeric α-syn in hippocampal rat primary neurons revealed an inhibitory impact of α-syn on SVs exocytosis, in a SNARE-dependent manner (Figure 1B) [77]. Precisely, by using a dynamin KO synapses model, a study showed that the monomeric form of α-syn is involved in regulating the early stages of synaptic vesicular endocytosis (Figure 1B) [78].

The widespread use of iPSC differentiated dopaminergic neurons has contributed to confirming the interactions between trafficking vesicles and the monomeric forms of α-syn. Recent work has reported that α-syn modulates the interaction of the hetero-oligomers composed of the scaffolding proteins Flotillin 1 and 2 (FLOT-1 and FLOT-2) which localize with endolysosomes [79]. In fact, monomeric and fibrillar forms of α-syn enhance the binding and clustering of the complex FLOT1-DAT (dopamine transporter) to the cell surface, resulting in improved endocytosis of the transporter protein, while downregulating its activity [79]. Interestingly, both FLOT1 and DAT have been reported to be integral components of LBs, suggesting a new pathway of α-syn trafficking [79].

### 3.2. α-Syn Aggregates Disrupt the Endolysosomal System

The process of α-syn aggregation involves a conformational change from the monomeric to the misfolded state to the oligomeric and fibrillar forms [80]. Interestingly, α-syn oligomeric forms have been reported to highly accumulate in PD brain patients and cause a dopaminergic neuronal loss in experimental animal models of PD [81,82].

Due to their high toxicity, the interactions between α-syn oligomers and trafficking vesicles appear to be of high relevance. It has been reported that misfolded forms of α-syn can get endocytosed and captured by endocytic vesicles of a 50 nm-vesicle lumen capacity limit [83]. Both the monomeric and oligomeric forms of α-syn appear to colocalize with specific markers of early endosomes (EEA1), recycling endosomes (Rab11), and late endosome/lysosomes (Lamp1, Lamp2a), resulting in the impairment of the autophagy-lysosomal pathway (Figure 1) [84]. Of note, oligomeric forms of α-syn have been reported to interact with the retromer pathway through Vacuolar Protein Sorting, VPS26 or VPS29, suggestive of defects in vesicles fission [18,85].

Moreover, the process of fibrillar α-syn internalization has been shown to occur via endocytosis (Figure 1), or through tunneling nanotubes, by following the neuronal endosomal pathway leading to lysosomal degradation [35,86]. One study showed that α-syn purified from human brain homogenates, added to H19-7 cells gets internalized in a Rab5-dependent manner, inducing the formation of LB-like intracytoplasmic inclusions [87]. Immunoblot and immunoprecipitation assays from patient brain samples suffering from Lewy-Body dementia, demonstrated that α-syn aggregates bind to Rab3a and disrupt Rab3a-rabphilin protein-protein interaction, resulting in an impairment of the docking of this complex to the synaptic vesicles transport (Figure 1B) [73]. With the growing interest in using preformed fibrils in neurons, Virginia Man-Yee Lee’s team has further shown that the fibrillar forms of α-syn can be internalized through the plasma membrane in all cell compartments [88]. These results were previously depicted using extracellular α-syn fibrils that were shown to be internalized by receptor-mediated endocytosis, which similarly suggest for the involvement of the lysosomal pathway in the degradation of internalized fibrils (Figure 1) [88]. In fact, studies using mammalian cell lines (COS-7, SH-SY5Y) and primary cortical neurons, revealed that α-syn fibrils can partially colocalize with the early endosomal marker EEA1 and the late endosomal/lysosomal marker Lamp2 (Figure 1A) [89]. More than just cellular uptake, exogenous amyloid fibrils have been shown to seed intracellular soluble proteins into insoluble inclusions in a co-culture model when overexpressing α-syn, with results demonstrating co-localization between α-syn and Rab5A/Rab7-positive vesicles [90]. Similarly in a study examining Rab5, co-localization between internalized fibrils and early endosomes was also reported in SH-SY5Y cells [91]. This study showed that fibrils appear to get acidified within hours of treatment before colocalizing with the late endosomal/lysosomal marker Lamp-1 [91]. Other studies focusing on the movement of TrkB receptors, known to undergo a retrograde axonal transport in endosomes, reported that the presence of abnormal α-syn fibrils in axons do not affect the overall mobility of late endosomes in axons [92,93]. More interestingly, the velocities of the late endosomes represented by the transport of GFP-Rab7 were shown to be impacted only retrogradely, suggestive of the trapping of TrkB receptors in late endosomes, preventing the redirection of vesicles to the lysosomal pathway (Figure 1) [93]. 

Likewise, several groups focused on examining the spreading of the pathogenic forms of α-syn in in vivo models. Using quantitative fluorescence microscopy, data revealed that in mouse neuron-like CAD cells, the majority (90%) of internalized α-syn fibrils localized within transport vesicles [94]. Moreover, in primary cortical neurons, part of the retrogradely internalized α-syn fibrils failed to overlap with the early endosomal markers EEA1 or Rab5 but rather colocalized with Rab7/Lamp1. This data suggested that the endosome/lysosome pathway is involved in the secretion of these proteins after their retrograde transport (Figure 1) [95]. Furthermore, it has been suggested that Rab5b-mediated endocytosis in neurons may be downregulated by other kinases such as LRRK2 [43]. LRRK2, a protein implicated in PD pathogenesis, and is also known to phosphorylate Rab5b and other 13 endogenous Rab proteins namely Rab3A/B/C/D, Rab5A/C, Rab8A/B, Rab10, Rab12, Rab29, Rab35, and Rab43 [69]. Additionally, in lamprey giant synapses, injection of aggregated α-syn showed to play a role in clathrin uncoating, evident by the accumulation of clathrin-coated-vesicles, proposing another impact of the fibrillar forms on early endocytosis (Figure 1B) [96].

Most of the current studies are focused on the interactions between fibrillar forms of α-syn and the trafficking vesicles, but these studies lack present information necessary for understanding the fate of cargo in these vesicles. In order to address this question, the predominant role of Vacuolar Protein Sorting 35 (VPS35) on the fate of fibrillar forms of α-syn has been emphasized by several groups [97,98,99,100,101,102]. In fact, mutations in VPS35 from the retromer complex, resulting in an enlargement of late endosomes, correlated with an increase in the aggregation of the oligomeric forms of α-syn [99]. More interestingly, in neuroblastoma cells, co-localization between Lamp-1 positive vesicles, and the aggregated phosphorylated fibrillary form of α-syn was described (Figure 1A) [99]. Likewise, VPS35 mutations in primary mutant dopaminergic neuronal mouse cells resulted in neuronal loss with impaired retrieval of LAMP2a to the lysosome, associated with a reduction of the degradation of the fibrillar forms of α-syn [102]. Furthermore, RNAi-mediated VPS35 knockdown (KD) has been shown to lead to the aberrant accumulation of insoluble α-syn in the lysosome, increasing the turnover of the mannose 6-phosphate receptor and subsequently, resulting in an impairment of the trafficking of its ligand, cathepsin D, to the lysosome [101].

As described above, endocytosed α-syn fibrils are encapsulated in the lumen of endocytic vesicles, and follow the endolysosomal pathway. Following the prion-like propagation, it appears that extracellular α-syn needs to directly interact with endogenous proteins in order to escape the classical endolysosomal pathway [35,86]. One explanation could be exploited by the functional impairment of the lysosomes. The lipid envelope of the endosomes ensures a robust structure, preventing content release in the cytoplasm, therefore the majority of the endocytosed α-syn gets degraded by lysosomes. However, the mechanism by which a fraction of α-syn fibrils escape this barrier, and initiate seeding remains unclear [35].

## 4. Interactions between α-Syn and the Degradation Vesicles

The main pathway responsible for α-syn degradation is lysosomal degradation, especially since α-syn accumulation has been associated with lysosomal inhibition [48]. Depending on α-syn’s structure and harbored mutations, it has been shown that α-syn can be degraded by both the CMA and macroautophagy pathways. The soluble monomeric forms of α-syn appear to be degraded by CMA, while its insoluble or pathological forms are most likely degraded by the macroautophagy pathway (Figure 1) [47].

The relationship between α-syn and CMA was first reported when using purified lysosomes. One study showed that wild-type monomeric, and mutant forms (A30P and A53T) of α-syn can be actively degraded by the CMA through interactions with LAMP2A [103]. Other α-syn mutant forms, such as dopamine-modified wild-type α-syn, also showed the propensity to inhibit the function of CMA leading to the activation of macroautophagy as a compensatory system, subsequently increasing toxicity in cells (Figure 1) [104,105].

### 4.1. Overexpression of the Monomeric Forms of α-Syn Inhibits Macroautophagy

It has been reported that, when monomeric forms of α-syn are overexpressed, macroautophagy is inhibited through the modulation of Rab1a activity, leading to the mislocalization of autophagy-related protein 9 (Atg9), a transmembrane protein normally expressed on autophagosomes [49]. In accordance, depletion of autophagy-related protein 7 (Atg7) in dopaminergic neurons, a protein involved in autophagosome assembly, appeared to lead to an increase in monomeric and aggregated forms of α-syn [106]. iPSC-derived astrocytes from PD patients harboring a LRRK2 GS2019S mutation, also tend to accumulate monomeric forms of α-syn, correlating with an impairment of both the CMA and macroautophagy pathways (Figure 1A) [76,107].

The CMA pathway is involved in the degradation of α-syn monomers and dimers through cathepsin D activity in lysosomes, while the inactivation of this protease is shown to lead to an increase in endogenous α-syn [104,108]. This interactive role was also reported with the inhibition of the CMA through the downregulation of LAMP2A, collectively leading to the formation of high molecular weight or detergent-insoluble oligomeric α-syn conformations [48].

Taken together, under its physiological state, α-syn seems to be a key interactor of the endolysosomal system, using its structure’s specificity to interact with all the different types of vesicles. As soluble monomeric forms, α-syn protein levels seem to be directed for lysosomal degradation through the ALP, the CMA, release through Rab11a-dependent recycling endosomal pathway, or through the exosome MVBs-dependent pathway (Figure 1).

### 4.2. α-Syn Aggregates Impair Lysosomal Functions

Misfolded forms of α-syn, such as oligomers, can also get degraded in lysosomes through cathepsin D; as the inhibition of lysosomes appears to lead to the accumulation of α-syn pathogenic forms in vitro and in the mouse brain [109,110]. Under general lysosomal inhibition, experiments performed in vitro and in vivo, shed light on the presence of specific co-localization between LAMP2A, LC3-II, and α-syn aggregates, confirming a strong involvement of the macroautophagy pathway in the degradation of the pathogenic forms of α-syn [111,112].

The direct link between cathepsin D activity and the accumulation of α-syn pathogenic forms was further characterized when accumulated α-syn pathogenic forms (oligomers and fibrils) were shown to lead to a decrease in endogenous cathepsin D activity, suggesting that the accumulation of α-syn forms may be the source of cathepsin D deficits (Figure 1) [84].

The degradation kinetics of fibrillary α-syn aggregates appear to be different depending on the cell type studied. For instance, in microglia, the degradation of internalized fibrillar aggregates appears to be more rapid in comparison to primary rat astrocytes and immortalized neuroblastoma cell lines such as SH-SY5Y [113]. Furthermore, when microglia is overactivated by lipopolysaccharide (LPS), an increase of α-syn aggregates in the cytoplasm is detected [113]. Similar to what has been reported with the oligomeric forms, the internalized fibrillar forms have been shown to localize within lysosomes and LAMP1 positive vesicles [94]. Interestingly, it was shown that in neurons, fibrils are capable of seeding aggregation of soluble α-syn, proposing a new pathway where overloaded lysosomes dispose aggregates by hijacking tunneling nanotubes-mediated intercellular trafficking [94,114]. Also, when lysosomes are pharmacologically impaired, the preformed fibrils persisted within the cell for days after their initial uptake, and have been found to be mostly trafficked to lysosomes leading to an acceleration of intracellular inclusion formation and recruitment [115]. Taken together, these data are consistent with the fact that α-syn aggregates are capable of impairing the functions of lysosomes, and thus increasing the risk of developing PD pathogenesis.

One of the elusive common questions of aggregate stability, is why aggregates appear to be persistently present after several days in culture? One explanation stems from the role of lysosomal dysfunction. Taguchi and collaborators elucidated on the exact relationship between the loss of GCase activity, glycosphingolipids (direct targets of GCase) accumulation, and α-syn aggregation [116]. Using in vitro and in vivo models of GBA mutation, they showed that glucosylsphingosine promotes the accumulation of pathological α-syn aggregates in GBA mutant and KO mice [116]. More interestingly, in human iPSCs dopaminergic neurons, the reduction of the accumulated glycosphingolipids resulted in a reduction in the pathological aggregated forms of α-syn, and this also showed to restore physiological α-syn conformers [117]. It appears very clear that GCase activity is crucial in modulating interactions between the fibrillar forms of α-syn and lysosomal vesicles. That is why modulators of GCase activity, like S-181, are now well studied with the aim to better examine possible applications to restore physiological degradation functions [118]. S-181 shows a propensity to increase GCase activity in human dopaminergic neurons harboring PD mutations (GBA1, LRRK2, and PARKIN), allowing for the restoration of lysosomal function, and reductions in the accumulation of oxidized dopamine and α-syn [118]. In this context, the SNARE protein ytk6 appears to also play a role in human midbrain neurons [119]. In fact, during lysosomal stress, ytk6 is physiologically active to promote hydrolase trafficking and enhance cellular clearance [119]. In the presence of α-syn aggregates, ytk6 is inactive when found bound to aggregates, thereby disabling the lysosomal stress response and facilitating protein accumulation. Thus, the proteolysis decline with the GCase activity within lysosomal compartments; suggests hydrolase miss trafficking which is upregulated by Rab1a [119,120,121].

Together, these results clearly highlight the major role played by the pathogenic forms of α-syn in disrupting lysosomal degradation, opening a door to emerging concepts that aim at better understanding these interactions over time. Such questions can include: When are the pathogenic forms capable of taking the lead on physiological degradation? And what is their trigger?

## 5. Conclusions

Converging lines of evidence specify for an interaction between the endolysosomal system and all forms of α-syn, starting from the early endosomes to the lysosomes where degradation happens. Using several cellular and animal models, major players of the endolysosomal system like Rab GTPase and LAMP proteins, retromer complex, hydrolases trafficking and GCase activity were shown to selectively interact with α-syn affecting its aggregation process (as reviewed in Table 1).

However, some questions are yet to be resolved, including questions on how α-syn changes its physiological interaction with the system? How α-syn takes over a system while integrating vesicles within their structure? Are there specific kinetics of association between α-syn aggregates and the endolysosomal vesicles? In order to best address these questions, there is an urgent need for more reliable PD models. In fact, current models of PD, based on α-syn overexpression, do not reproduce all characteristics of α-syn accumulation and LBs formation in PD patients’ brains. This major lack of proper PD models, makes the analysis of the interactions between LBs and the endolysosomal system very difficult. Moreover, the results appear to be highly variable depending on the fibrillar forms examined. The differences observed in terms of kinetics uptake and fibrils degradation may be linked to the differences in culture conditions and preparation of α-syn assemblies [122,123]. In fact, a recent study proposed a gold standard method for the preparation of monomers, oligomers, and fibrillar forms of α-syn to increase the reproducibility of the results [122]. To date, the only published model recapitulating almost all the cardinal features, which is based on preformed fibrils uptake, requires several days of culture and does not give the spatiotemporal resolution needed to study such mechanisms [9].

To overcome this lack of spatiotemporality, recent work focused on making use of the properties of the Arabidopsis thaliana cryptochrome 2 (CRY2) protein, which has the capacity to oligomerizes quickly and reversibly in the presence of blue light, showed to give very high spatiotemporal resolution for α-syn, tau or amyloid-β aggregate formation [124,125,126]. These methods could allow us to answer unresolved questions related to early key events in PD pathogenesis such as the interactions with the endolysosomal system, which can further allow us to find potential new disease-modifying drugs that can slow or stop the underlying neurodegenerative process.

## Figures and Tables

**Figure 1 biomolecules-11-01333-f001:**
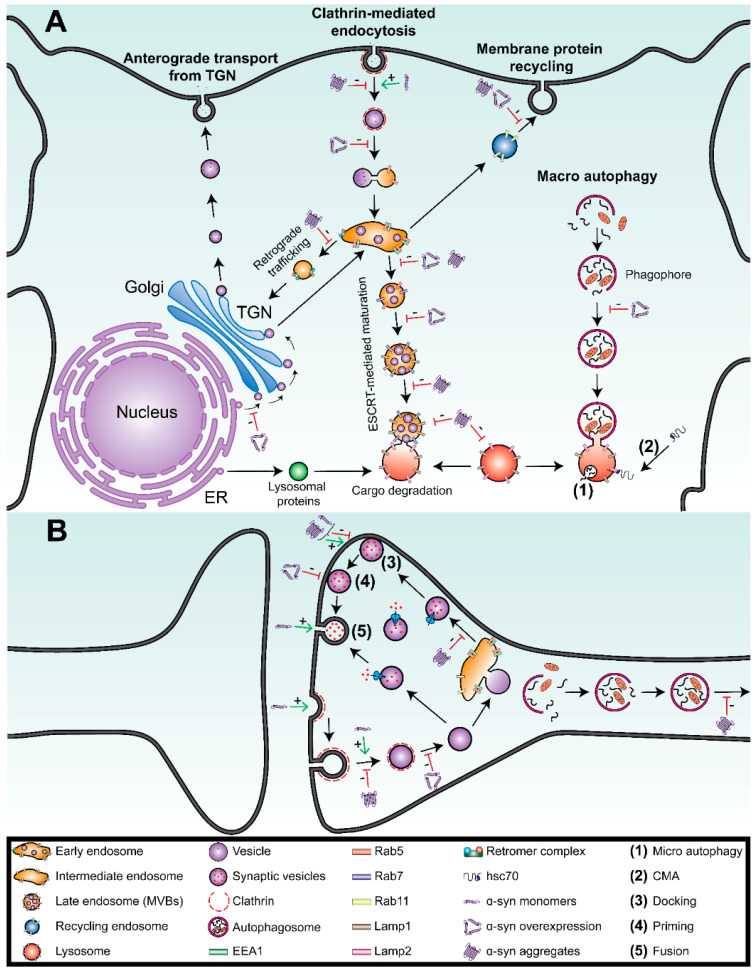
The complex relationship between alpha-synuclein and the endolysosomal system in Parkinson’s disease. (**A**) In the soma, essential functional proteins are produced in the endoplasmic reticulum (ER, purple) and are post-translationally modified in the Golgi apparatus (Golgi, blue). The overexpression of monomeric α-syn inhibits this trafficking process. Proteins, like lysosomal hydrolases, are packed into vesicles (purple) and delivered to the *trans*-Golgi network (TGN, blue). From the TGN, depending on the post-translational modifications, the vesicles are directed towards early endosomes (orange) for further degradation in the lysosomes, or to the secretory pathway (anterograde transport from the TGN). Simultaneously, the extracellular vesicles are endocytosed following the canonical endosomal pathway initiated by the clathrin-mediated endocytosis (CME, red dashes)—which can be upregulated by monomeric α-syn and downregulated by the aggregated forms—followed by vesicle uncoating (inhibited by α-syn overexpression) and fusion with early endosomes (orange). These early endosomes, which are mildly acidic, act as a sorting station, targeting cargo towards multiple destinations. With the help of the endosomal sorting complex required for transport (ESCRT), early endosomes mature into late endosomes where luminal pH decreases, while sequestrating ubiquitinylated proteins into intraluminal vesicles forming multivesicular bodies (MVBs) before fusing with lysosomes to initiate the proteolytic degradation of the cargo. Glucocerebrosidase (*GBA*/GCase) and Cathepsin D are crucial for normal lysosomal function, that is negatively impacted by the presence of aggregated pathological forms of α-syn. The endosomal maturation pathway is inhibited at several steps by these two forms of α-syn. Early endosomes that are not sorted towards cargo degradation pathway follow the recycling pathway, and back to the plasma membrane (blue) where these two forms of α-syn also play an inhibitory role. Early endosomes may also undergo retrograde transport to the TGN through the retromer complex where Vacuolar Protein Sorting 35 (VPS35) takes part, which is negatively mediated by the presence of the aggregated forms of α-syn. Cargo in the phagocytic pathway, mainly composed of misfolded proteins and dysfunctional organelles, is degraded through the endolysosomal pathway when double-membraned phagosomes fuse with late endosomes or lysosomes. The complete process is called autophagy, and is dissected into three pathways: Macroautophagy, microautophagy (**1**), and chaperone-mediated autophagy (**2**). Overexpression of α-syn is known to inhibit phagophore maturation. (**B**) At the synapse, neurotransmitters are released following the synaptic vesicle (SV) cycle, a key part of neuronal physiology. The synaptic exocytosis is divided into three sequential steps: docking (**3**), priming (**4**), and vesicle fusion (**5**). After docking, the vesicles are primed through a series of ATP-dependent reactions, and are then released after an action potential that triggers the calcium influx allowing for fusion pore opening. α-Syn plays a major role in these 3 steps depending on its conformation: monomeric and aggregated forms may upregulate or downregulate the docking of SVs, overexpressed monomeric forms downregulate the priming of SVs while monomeric forms upregulate the fusion. Empty SVs are further recycled through CME. Clathrin quickly forms a layer around invaginations and induces membrane fission with the help of dynamins and the monomeric forms of α-syn that play a role in membrane binding, inducing SV curvature formation. While on the contrary, aggregated forms of α-syn inhibit SVs fission. The endosomes derived from the plasma membrane also form SVs. Synaptic autophagy is also crucial in maintaining the health of the pre-synaptic terminal, which may be altered by the overexpressed monomeric forms of α-syn. In the axon, endosomes and autophagosomes are retrogradely transported from the synaptic terminals to the soma.

**Table 1 biomolecules-11-01333-t001:** Interactions between alpha-synuclein and main vesicular proteins.

Alpha-Synuclein Conformation	Vesicular Proteins	Evidence of Interaction	Substrates	**References**
Monomeric forms	Rab5	Colocalization	SH-SY5Y, 293T	[76]
	EEA1	Colocalization	H4, primary cortical neurons (rat)	[84]
	EEA1	Colocalization	293T, dopaminergic neurons (human iPSC)	[79]
	Rab11	Colocalization	H4, primary cortical neurons (rat)	[84]
	Rab11	Colocalization	293T, dopaminergic neurons (human iPSC)	[79]
	Rab11	Colocalization	SH-SY5Y, 293T	[76]
	Rab7	Colocalization	SH-SY5Y, 293T	[76]
	Rab7	Colocalization	293T, dopaminergic neurons (human iPSC)	[79]
	Lamp1	Colocalization	H4, primary cortical neurons (rat)	[84]
	Lamp1	Colocalization	SH-SY5Y, 293T	[76]
	Lamp2a	Co-IP	PC-12	[103]
	Lamp2a	Colocalization	Astrocytes (human iPSC)	[107]
	Lamp2a	Colocalization	H4, primary cortical neurons (rat)	[84]
Aggregated forms	Rab5	Colocalization	SH-SY5Y	[90]
	Rab5	Colocalization	SH-SY5Y, KG1C	[91]
	Rab5	Colocalization	Primary hippocampal neurons (mice)	[127]
	EEA1	Colocalization	Mouse neuron-like CAD cells	[94]
	EEA1	Colocalization	H4, primary cortical neurons (rat)	[84]
	EEA1	Colocalization	293T, dopaminergic neurons (human iPSC)	[79]
	EEA1	Colocalization	COS-7, SH-SY5Y, primary cortical neurons (rat)	[110]
	Rab11	Colocalization, Co-IP	MES	[128]
	Rab11	Colocalization	293T, dopaminergic neurons (human iPSC)	[79]
	Rab11	Colocalization, Co-IP	SH-SY5Y, brain rat homogenates	[72]
	Rab11	Colocalization	H4, primary cortical neurons (rat)	[84]
	Rab7	Colocalization	293T, dopaminergic neurons (human iPSC)	[79]
	Rab7	Colocalization	SH-SY5Y	[90]
	Rab7	Colocalization	Primary hippocampal neurons (mice)	[127]
	Lamp1	Colocalization	H4, transgenic mice, LBD brain patients	[111]
	Lamp1	Colocalization	Mouse neuron-like CAD cells	[94]
	Lamp1	Colocalization	Primary cortical neurons (mice)	[95]
	Lamp1	Colocalization	SH-SY5Y	[99]
	Lamp1	Colocalization	H4, primary cortical neurons (rat)	[84]
	Lamp1	Colocalization	SH-SY5Y, KG1C	[91]
	Lamp2a	Colocalization	H4, primary cortical neurons (rat)	[84]
	Lamp2a	Colocalization	H4, transgenic mice, LBD brain patients	[111]
	Lamp2a	Colocalization	H4, transgenic mice, LBD brain patients	[111]
	Lamp2a	Colocalization	COS-7, SH-SY5Y, primary cortical neurons (rat)	[110]
	Rab3a	Co-IP	LBD brain patients	[73]
	VPS26	Proximity labeling	Primary cortical neurons (rat)	[85]
	VPS29	Proximity labeling	Primary cortical neurons (rat)	[85]
	CD63	Colocalization	Primary hippocampal neurons (mice)	[127]
	CD83	Colocalization	Primary hippocampal neurons (mice)	[127]
	Ykt6	Pull down	Midbrain neurons (human iPSC)	[119]

## Data Availability

No new data were created or analyzed in this study. Data sharing is not applicable to this article.

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
