# Peer review of "Alpha-Synuclein and the Endolysosomal System in Parkinson’s Disease: Guilty by Association"

_biomolecules, 2021, doi:10.3390/biom11091333_

Round 1
Reviewer 1 Report
In this review entitled “Alpha-synuclein and the endolysosomal system in Parkinson’s 2 disease: guilty by association” Maxime Teixeira et al. have analysed the different experimental evidence connecting alpha-synuclein to endo-lysosomal system in Parkinson’s disease. The topic is not new and already well summarized in other reviews but the manuscript is well written and captivating.
Comments:
- In Figure 1 put A and then B for an easier reading.
Reviewer 2 Report
This review manuscript by Teixeira and colleagues provides a comprehensive overview on the interactions between α-synuclein aggregation and the endolysosomal system. As the authors point out, there is accumulating evidence that suggests the involvement of the endolysosomal system in PD pathogenesis. Therefore it is extremely timely to publish this review. This reviewer do not have major concerns on the submitted version of the manuscript and just give some minor comments that might be useful to further improve it.
Line 35, “non-amyloidic (NAC)” is not correct. NAC stands for “non-Aβ component of AD amyloid” (Ueda et al., PNAS, 1993).
Lines 130–134, it is a little bit confusing to use (1)(2)(3) as the same numbering is used in Figure 1A. Consider use (i)(ii) or (a)(b) etc.
Line 141, “key Rab regulators” can simply be “key regulators”.
Line 207, “substrates” is not commonly used for Rab proteins. “Effectors” and/or “regulators” might be more suitable for this context.
Lines 300–302, ref 69 does not show physical interactions of LRRK2 with 13 Rab proteins. If the intension here is a kinase-substrate interaction, the word “interaction” is somewhat misleading.
Figure 1, I think that the panel A should be on top of the panel B, if there is no special reason for swapping A and B.
